# Stress Salivary Biomarkers Variation during the Work Day in Emergencies in Healthcare Professionals

**DOI:** 10.3390/ijerph18083937

**Published:** 2021-04-09

**Authors:** Daniel Pérez-Valdecantos, Alberto Caballero-García, Teodosia Del Castillo-Sanz, Hugo J. Bello, Enrique Roche, Alfredo Córdova

**Affiliations:** 1Departamento de Bioquímica, Biología Molecular y Fisiología, Facultad de Ciencias de la Salud, GIR de Ejercicio Físico y Envejecimiento, Campus Universitario “Los Pajaritos”, Universidad de Valladolid, 42004 Soria, Spain; danielperezvaldecantos@gmail.com; 2Departamento de Anatomía y Radiología, Facultad de Ciencias de la Salud, GIR de Ejercicio Físico y Envejecimiento, Campus Universitario “Los Pajaritos”, Universidad de Valladolid, 42004 Soria, Spain; alberto.caballero@uva.es; 3Gerencia de Emergencias Sanitarias de Castilla y León, UME Soria, Hospital Virgen del Mirón, 42005 Soria, Spain; tcastillosanz@gmail.com; 4Departamento de Matemáticas, Escuela de Ingeniería de la Industria Forestal, Agronómica y de la Bioenergía, Campus Universitario “Los Pajaritos”, Universidad de Valladolid, 42004 Soria, Spain; hjbello.wk@gmail.com; 5Instituto de Bioingeniería y Departamento de Biología Aplicada-Nutrición, Universidad Miguel Hernández, 03202 Elche, Spain; eroche@umh.es; 6Instituto de Investigación Sanitaria y Biomédica de Alicante (ISABIAL), 03010 Alicante, Spain; 7CIBER Fisiopatología de la Obesidad y Nutrición (CIBEROBN), Instituto de Salud Carlos III (ISCIII), 28029 Madrid, Spain

**Keywords:** amylase, cortisol, DHEA, emergencies, health professionals, stress

## Abstract

Objective: The work of health professionals in hospital emergency rooms is highly demanding due to the decisions they must take. In the present study, we consider assessing stress response in emergency health workers, measuring related biomarkers such as cortisol, dehydroepiandrosterone (DHEA) and salivary α-amylase during the whole working day. Method: An analytical, descriptive and cross-sectional study was carried out. The study was conducted in the emergency rooms of two public hospitals. Ninety-seven professionals participated, 45 corresponding to one hospital and 52 to the other. Four salivary samples were obtained according to circadian rhythms: at 8:00, 12:00, 15:00 and 00:00 h/24 h. The data were subsequently analyzed. Results: Cortisol levels decreased throughout the working day, with minimum values being at 24 h. A similar pattern was observed in DHEA. The α-amylase levels increased throughout the working day, reaching its peak at 15:00 h, and decreasing at 24 h, compared to the data from the rest of the working day. Conclusions: Since reference/baseline values are not presented, this work is focused on a stress situation experienced during one regular working day in emergency rooms with no extreme situations. In this context, stress, measured through cortisol and α-amylase, is present in emergency room doctors and nurses. However, the increase in DHEA, due to its anabolic condition, could counteract their effect, suggesting a positive effect on their professional actions.

## 1. Introduction

The work of healthcare professionals in emergency medical services is associated with chronic exposure to daily work stress circumstances, which, together with regular daily events, affect stress biomarkers [1]. This situation may be capable of overloading stress reaction adaptation capacity and such individuals can become vulnerable to illness.

Seyle [2] defined stress as “a coordinated set of physiological reactions to any form of harmful stimulus.” Seyle developed a multi-stress model that included the characteristics of the body’s physiological response to the stressful demands, as well as the harmful organic consequences produced due to an excessive or prolonged exposure to the stress situation [2]. Other authors [3] studied the impact of stress, which were classified into five areas: physiological effects, effects on task execution, effects on interpersonal and affective behaviors, effects on verbal and nonverbal behavior, and effects on adaptation processes.

In physiological circumstances, i.e., when an individual is not exposed to stress situations, hormonal secretion is regulated by the circadian rhythm. Cortisol concentrations are high in the morning, peaking thirty minutes after awakening, and progressively dropping throughout the day to lower concentrations in the early evening. This typical secretion pattern is crucial to the functions of all other systems of the human organism [4].

The immediate stress reaction, or the “fight-or-flight response”, involves a rapid activation of the adrenal medulla through the autonomous nervous system (ANS), leading to high concentrations of circulating epinephrine and norepinephrine. However, in the long term, the coordinated stress response involves the participation of the hypothalamic-pituitary-adrenal (HPA) axis and the adrenomedullar sympathetic system (AMSS). The response is validated according to the levels of adrenocorticotropic hormone (ACTH) that initiates the process, as well as the levels of cortisol and catecholamines (epinephrine and norepinephrine) [4]. Therefore, chronic stress is associated with the activation of the HPA axis, resulting in increased plasmatic and salivary cortisol levels. Cortisol levels in saliva are closely related to blood cortisol, reliably reflecting the HPA axis’ activity [5,6]. In this vein, acute stress is associated to the activation of the AMSS, resulting in epinephrine and norepinephrine increases that lead to salivary α-amylase rise [7]. As can be seen, all these biomarkers can be determined in blood and saliva. Since the puncture to obtain the blood sample can produce some stress by itself, it seems advisable to use other methods whose collection is not invasive and does not generate stress, such as saliva. In addition, saliva is also very easy to obtain by staff with minimal training [5,6]. Salivary concentrations of fat-soluble or non-soluble steroids represent a percentage of their plasma concentration [8]. Nevertheless, saliva allows for the study of steroidal hormones in their free fraction, with the advantage of easy sample collection.

In the alarm phase described by Seyle [2], the HPA axis stimulation increases the production of cortisol, preparing the body for a sustained response to stress. The body also synthesizes DHEA and DHEA-sulfate (DHEA-S) to compensate for the harmful effects of cortisol. DHEA and DHEA-S are precursors of secreted androgens in response to ACTH [9]. DHEA and DHEA-S have been shown to have neuroprotective, antioxidant, anti-inflammatory and anti-glucocorticoid effects [10,11]. In addition, DHEA can reach the saliva by intracellular passive diffusion, allowing for an easy determination.

The work of health professionals in hospital emergency rooms is of high psychological demand due to the decisions they must make. High and sustained levels of stress can increase the risk of cardiovascular disease, as well as increased susceptibility to infections and mental disorders, affecting the task performance of health professionals [9,12]. In this context, cortisol is the parameter mostly accepted by the scientific community as the best marker for stress. Likewise, norepinephrine (noradrenaline) testosterone, dehydroepiandrosterone (DHEA) and α-amylase are also considered to be complementary stress markers [13]. However, several key factors have to be taken into account regarding these stress markers.

The emergency department is a stressful workplace with excessive workloads or high demands on patient care, time pressures and the intensive use of sophisticated technologies. In certain occasions, the service suffers increased patient inflow and reduced capacity for patient care managerial skill demands on the healthcare staff [14]. Prolonged emotional pressure or chronic stress can lead to a broad spectrum of physical and psychological diseases [15,16]. The perception of a distressing experience depends mainly on individual aspects and is physiologically difficult to measure. The European Union (EU) indicates that stress associated with the workplace is the second most common work problem after musculoskeletal disorders. EU defines stress as “a set of neuroendocrine, immunological and emotional processes and responses”.

In the context of the present report, several studies have been published regarding the response to stress of health professionals in emergency rooms. However, the results of these studies were contradictory. For example, some studies noticed that, during a shift in the emergency room, there is an increase in salivary cortisol levels [17,18]. However, other authors did not observe such cortisol elevations [19]. However, regarding the long-term response during the whole working day in terms of psychological aspects, an increase in anxiety, depression and chronic stress has been described in emergency room nurses and medicine professionals [20]. In view of these data, we considered assessing in the present report the stress response of emergency health professionals throughout the whole day (working and in daily activities outside the working place), determining variations in the levels of salivary cortisol and α-amylase, as well as the counteracting effects of DHEA.

## 2. Material and Methods

An analytical, descriptive and cross-sectional study was carried out during the months of July (recruitment) and August (determinations) 2019 in the emergency rooms of two public hospitals: Hospital Clínico Universitario de Valladolid (HCUV) (third level) and Hospital Santa Bárbara de Soria (HSBS) (second level). The project was approved by the Ethics Committee of Universidad de Valladolid (Ref. CEIC 1984).

The total sample of the study was 97 participants: 59 were certified nurses (10 men and 49 women) and 38 were medical doctors (10 men and 28 women). Regarding distribution in both hospitals, 45 professionals were from HCUV and 52 from HSBS. Regarding task distribution, 66 worked in morning (8:00–15:00 h) shifts and 31 on call during the afternoon (15:00–22:00 h). Regarding employment situation, 27 professionals were permanent staff (around 15 years of experience), 34 were temporary substitutes (around 4 years of experience), 20 interims (around 12 years of experience) and 14 were training professionals (MIR category) (around 3–4 years of experience) (Table 1). All participants had stable lifestyle and family habits according to their responsibilities. This aspect was taken into account for the study, as it could possibly influence stress response. 

A total of 105 professionals worked in the emergency rooms of both hospitals. The application of the exclusion criteria (see below) resulted in a final *n* = 97 participants. Selected subjects completed a short demographic questionnaire, an on-the-job behaviour inventory, and the revised version of the Medical Personnel Stress Survey (MPSS-R). The MPSS-R is a 40-item questionnaire with ten items on each of four subscales: somatic distress, negative patient attitudes, job dissatisfaction, and organizational stress. A total stress score may be calculated as the sum of these components. A total abbreviated MPSS-R score of >50 is considered to reflect high levels of occupational stress [21].

All selected participants were healthy, with no mental or physical pathology that could hamper their work. The exclusion criteria were: medical leave of absence for a period in excess of fifteen days over the preceding thirty days; smokers or history or smoking over the last five years; abusive use of alcoholic beverages or prior history over the last five years; use of medications that influence the HPA axis (glucocorticoids, steroids, beta-blockers, antidepressants, melatonin, or any other psychoactive drugs); use of glucocorticoids over the last three months; medically diagnosed neurological or psychiatric illness; night shift working activity in another institution. None of them suffered from endocrine-type pathologies that could alter the endocrine stress response. Salivary measurements of cortisol, α-amylase and DHEA were used to assess the stress response. Prior to the study, each participant recorded their daily sleep log (bedtime and waking time) during the previous five days before the test. This was performed to verify the existence of a regular resting schedule that could alter stress parameters. Participants in the study reported a constant resting schedule.

Saliva samples were obtained using the Salivette commercial kit^®^ (Sarstedt International, Nombrecht, Germany). To have the saliva collection, participants were advised to avoid eating or smoking the 60 min before collecting each sample. Once collected, samples were maintained on ice first and then at −20 °C until analysis. At the laboratory, samples were thaw, centrifuged at 3000 rpm for 5 min at 4 °C and analyzed according to manufacturer instructions. Subsequent detection was performed by Elisa immunoassay for cortisol (SALV-2930 DRG, Marburg, Germany), α-amylase (EIA-5836 DRG, Marburg, Germany) and DHEA (SLV3012 DRG, Marburg, Germany). The reference values were set according to the bibliography and manufacturer’s specifications. The samples were obtained by taking into account circadian rhythms at four moments of the day: 8:00, 12:00, 15:00 and 00:00 h/24 h. The days for saliva collection were regular working days with no extreme or particular emergency events, such as experienced in 2020 during COVID pandemic situation. Saliva collection was interrupted in cases that a sudden event occurred. Participants knew the schedule of the intervention in advance and thereby the day planned to obtain the corresponding saliva samples. In addition, the day of saliva collection by a specialized technician, each participant was notified by a cell-phone call 30 min before obtaining each sample at work or at home, depending of the working shift. Only saliva sample at 24 h was obtained by the participant itself, according to instructions provided by the technician and maintained at −20 °C until analysis at the hospital laboratory.

Statistical analysis: The data were analyzed with R, R-Studio, and Python software package (Pandas, Numpy). Results were expressed as the mean ± standard deviation (SD) and 95% confidence interval. Confidence intervals were obtained using non-parametric bootstrap analysis. A hypothesis test analysis was performed to determine the statistical evidence for the decreases of cortisol and DHEA, and the increase of α-amylase. Variables did not follow a normal distribution; therefore, non-parametric methods were mainly used. The Kruskall-Wallis test was performed to determine significant differences (*p* < 0.05) in the concentrations of cortisol, α-amylase and DHEA at the different times. The Wilcoxon signed-rank test indicated significantly (*p* < 0.05) that DHEA and α-amylase levels at 8:00 h were lower than at 24 h, and cortisol levels were higher at 8:00 h than at 24 h.

## 3. Results

Table 2 shows the total values obtained from the three stress-related biomarkers determined in this study. First of all, all the subjects presented high cortisol levels, taking into account the reference ranges provided by the clinical analysis laboratories of both hospitals. In addition, cortisol decreased significantly throughout the working day. A similar pattern was observed for DHEA. However, α-amylase was increased throughout the working day, reaching a maximum at 15:00 h in the whole population sample, in each hospital, in medical doctors, in nurses, in the day shift as well as in the afternoon shift (Table 2). The elevation of the DHEA/cortisol ratio may constitute an important element in response to stress. In our data (Table 2), we observed that this relationship also increased throughout the working day, indicating a predominance of anabolic processes. When this ratio decreases as a reflection of catabolism predominance, this is associated with an increased risk of suffering from cardiovascular diseases or metabolic syndrome, among others [22,23,24]. In addition, when both hospitals were compared (HCUV vs. HSBS), only α-amylase showed significant differences at 8:00 and 12:00 h (Table 2). Furthermore, when medical doctors were compared with nurses, significant differences were found in α-amylase levels at 12:00 and 15:00 h (Table 2). Finally, when the day shift was compared with the afternoon shift, significant differences were found in DHEA and α-amylase at 12:00 and 15:00 h (Table 2).

Table 3 shows the results corresponding to the situation of perceived stress according to the MPSS-R questionnaire. The mean value of the total score was 59.5 ± 5.8 (HSBS) and 71.7 ± 5.6 (HCUV). Measured stress levels were high for all the groups studied, particularly for HCUV. Somatic distress and organizational stress were the most prominent markers of stress, followed by job dissatisfaction and the negative attitudes towards patients. Nevertheless, MPSS-R is a subjective questionnaire and no correlations have been found, as compared to variations in salivary biomarkers.

Figure 1 depicts the cortisol and DHEA levels, according to gender. The pattern of both hormones was similar in both genders, decreasing significantly throughout the working day.

Figure 2 shows the results, differentiated by sex, of α-amylase levels throughout the working day. A similar pattern was observed in both genders, increasing throughout the day, reaching a peak at 15:00 h and subsequently decreasing at 24 h.

Figure 3 shows the correlation chart between the three biomarkers. The only positive correlation between biomarkers (*r* = 0.5) appeared between cortisol levels at 24 h and α-amylase at 8:00 h.

## 4. Discussion

It is known that both acute and chronic stress imply alterations in the adrenal axis. This study demonstrated that the excretion of salivary biomarkers could change due to stress. The main observation of the present report indicates that the stress generated throughout the whole working day relies in the AMSS response. Therefore, we hypothesize that the adrenal cortex response could be adapted to the daily situations experienced by hospital emergency care professionals. In this context, the data obtained could be considered representative of a chronic stress situation. This assumption could be considered for regular working days with no particular/extreme events that could alter the emergency care routine, as shown in this report. These days are usually the main part of the time in emergency rooms. However, the experimental design indicates that we are determining acute stress, because only one particular day was taken into account. In addition, the biochemical markers of stress, salivary α-amylase and cortisol, seemed to have different reaction profiles, as previously confirmed by other studies [25]. Additional research that takes longer periods of time needs to be performed in order to address these questions.

Regarding cortisol, this is a corticosteroid hormone that influences memory consolidation in humans. Its activation and recovery is slower than α-amylase [26]. Our data are in accordance with this difference. One previous study performed a hospital-emergency simulation [6], reporting increases in α-amylase, similar to those observed in a real work scenario (morning call in the present report). The same authors found no variations of cortisol, contrary to the observations of the present study. However, our results are consistent with other simulation studies that have observed a significant reduction in cortisol values over time [27]. Altogether, this suggests that, despite the pattern followed by α-amylase, the cortisol pattern indicates a tendency to decrease the accumulated stress during the working days in health professionals [27]. Certain authors indicate that increased salivary amylase levels may be considered as a faster reaction to stress than cortisol, suggesting that it is a better stress indicator. In this context, salivary α-amylase seems to be more sensitive than cortisol as an indicator of an adaptation to the stress situation rather than an ANS dysregulation [26,28]. Further research is necessary to answer this question. In this context, a decrease in α-amylase production could be related to a decrease in the stress indicating a calming or relaxation situation. Our hypothesis is that the response of the suprarenal medulla, from which α-amylase depends on, is faster than the cortex response, from which cortisol depends, in accordance with [28]. In addition, the accommodation to the situation favors the decrease of cortisol levels [26]. In this sense, Valentin et al. [6] found a response similar to that observed in our study, suggesting that, despite the results of α-amylase, perhaps there was a decrease in stress over time, indicating a possible adaptation throughout the workday. Work stress was observed by Bedini et al. [29] in emergency phone operators. In this report, salivary cortisol levels increased at reception of incoming calls as the most stressful situation. This rise depends on the perceived stress and severity of the emergency call. Then cortisol levels decrease when decisions were taken and there is not more contact with callers [29]. Although the study design is not similar to the situation managed in our study, we agree that a chronic exposure to stress, often experienced by emergency professionals, can lead to hyporeactivity in cortisol response. Altogether, the evidence indicates that α-amylase seems to be a consistent marker in situations of acute stress with instrumental applications in particular cases, such as in the present study performed in emergency health professionals [7].

Regarding variations in DHEA, their levels decreased over the study period of one day. The results coincide with other studies where the participants, under prolonged stress, lead to exhaustion [30]. As previously indicated, DHEA seems to play a protective role during acute stress as an antagonist to the effects of cortisol [11]. The results of the present report go in this direction, indicating that DHEA levels seem to counteract cortisol increases. We hypothesize that the situation observed in the pattern of the different markers could have a positive interpretation. We understand that the patient care situation in emergency rooms generates alertness in healthcare professionals. However, the stress at the working place is accompanied by the stress associated with regular daily activities (shopping, driving, cooking, etc.). According to the α-amylase level increases in the afternoon call staff, we suggest that the daily activities aside from the workplace may also contribute to the acute stress experienced by participants. The accumulated stress reflected by the response of cortisol, decreases throughout the working day. This could be a positive element that in turn would lead to better performance of the work functions in both the nursing and medical sections. In this vein, DeMaria et al. [31] investigated the addition of emotional stressors in simulated cardiopulmonary stop training. Participants were able to remember the events of scenarios in which they felt they had failed, demonstrating that emotional stress can improve the stages of memory, the creation of new memories or the persistence of memories, as well as the ability to remember these memories.

The complexity of responses involving stress markers may reflect a variety of factors that influence measurements, such as: different time of day for experiments, eating interference on collected saliva and individual stress perception [32]. While this is true, our study demonstrates a poor correlation among the different acute stress markers analyzed (Figure 3). A positive correlation (*r* = 0.5) appeared between cortisol levels at 24 h and α-amylase at 8:00 h. A likely explanation is that the decrease in cortisol at the end of the day, prepares for an increased response of α-amylase that reacts quicker than cortisol in response to stress.

However, as has been previously published [33], these results may be explained by recent stress theories that, using the concept of allostasis, explain stress and coping strategies as an integrative state determined by genetic, developmental, environmental, and previous experiential factors. According to the allostatic concept of stress, adaptive activities of effector systems are coordinated in specific patterns.

It is well recognized that healthcare professional staff present chronic work-related stress, which can manifest in altered stress biomarker concentrations that can affect individuals as well as healthcare team performances when treating patients. In this way, for example, medical students with poor strategies to face stress show reduced laparoscopic skills [34]. Over time, stress leads to observable changes, indicating the existence of a relationship between acute stress and established performance [35,36]. We have observed that there is a high level of stress in the healthcare personnel through the answers in the MPSS-R questionnaire. This stress appears to manifest itself either through negative organizational attitudes or through patient care dimensions. However, the personnel do not manifest fatigue, sickness, or other psychophysiological markers of stress. Therefore, it is not easy to evidence that they are stressed, making it difficult to interpret their role in emergency rooms. By this, emergency medical services should acknowledge differences in stress levels between differing organizations and between individuals within the same organization, so that they may adjust their stress management priorities effectively.

However, several studies have been published that look at the different responses to stress depending on gender. Takai et al. [37] found no gender differences in the response of cortisol and α-amylase, which coincides with the present report. However, Kirschbaum et al. [38] found differences that they attributed to the psychological profile of the subjects studied, noting that the observed differences were due to the participants presenting depressive mood and low self-esteem. In this context, the data presented from our research seems to be relevant, since the study involved all healthcare professionals working in the emergency rooms of two hospitals, thus avoiding any bias. However, other studies revisited present biases regarding the selection of participants. Therefore, although their conclusions are interesting, they cannot be compared to the data provided in this work. In this sense, Ruotsalainen et al. [39] indicates that, in response to the ever-changing demands of their work function, healthcare professionals could learn to mitigate their reactions, and subsequently the negative consequences in situations of acute stress.

One of the limitations of this study may be that the medical doctors and nurses were studied over 1 day of work, and therefore, were only exposed to the changes in the stress hormones secretions during this particular day. However, taking into account that they worked 8-h shifts regularly, the study findings might be representative of all regular working days. Notwithstanding, it is clear that carrying out a study of much longer duration could provide better evidence of the pattern of hormones related to stress. One strong aspect of the study is that a specialized technician mainly collected the samples and it would therefore not affect either the motivation or the accuracy of the collection time.

## 5. Conclusions

In conclusion, salivary α-amylase appears to be a more sensitive marker than salivary cortisol to detect stress in a whole day environment, but this does not condition the overall coping response of healthcare professionals. This study showed that stress in teams of doctors and nurses is a real fact. However, from a practical point of view, the increase in DHEA seems to have a positive effect on these professionals, making them more capable and resolute in their actions.

## Figures and Tables

**Figure 1 ijerph-18-03937-f001:**
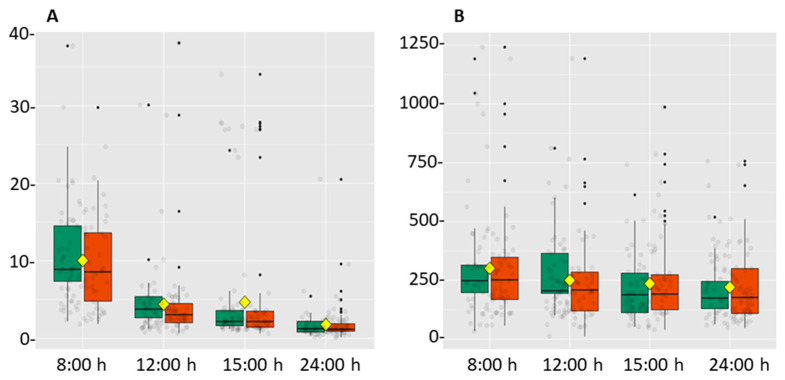
Box diagram of the pattern followed by salivary cortisol (**A**) and DHEA (**B**) throughout the working day according to gender. Green boxes correspond to men and red boxes to women. Hormone levels are represented by arbitrary values. The yellow square corresponds to the mean value, taking into account both genders. Black and grey dots represent the data distribution.

**Figure 2 ijerph-18-03937-f002:**
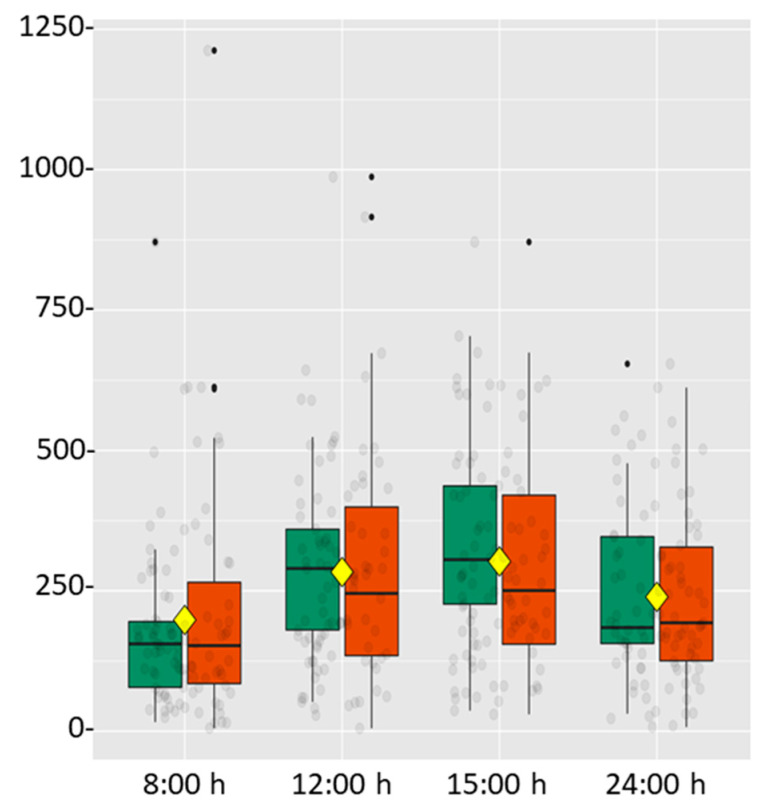
Box diagram of the pattern followed by salivary α-amylase throughout the working day according to gender. Green boxes correspond to men and red boxes to women. Enzyme levels are represented by arbitrary values. The yellow square corresponds to the mean value taking into account both genders. Black and grey dots represent the data distribution.

**Figure 3 ijerph-18-03937-f003:**
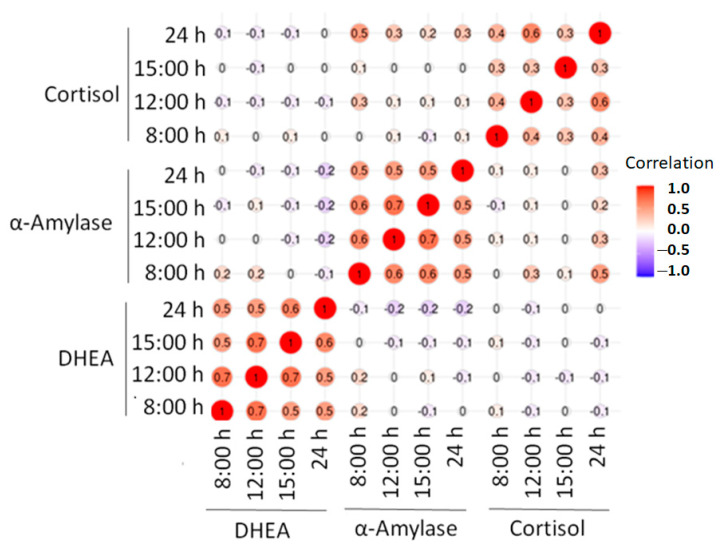
Correlation chart of the 3 salivary biomarkers (cortisol, α-amylase and DHEA) at different moments of the day (8:00 h, 12:00 h, 15:00 h and 24 h).

**Table 1 ijerph-18-03937-t001:** Group numbers and percentages.

Professional	*n*	%
Total	97	20.6/79.4 (Men/Women)
Nurses	59	16.9/83.1 (Men/Women)
Medical Doctors	38	26.3/73.7 (Men/Women)
HCUV	45	46.4
HSBS	52	53.6

Abbreviations used: HCUV, Hospital Clínico Universitario de Valladolid; HSBS, Hospital Santa Bárbara de Soria.

**Table 2 ijerph-18-03937-t002:** 95% confidence intervals for the mean, calculated using bootstrap for normalization. Measurements of variations (mean + SD) throughout the working day of salivary cortisol, α-amylase and DHEA.

Biomarker (Units)	8:00 h	12:00 h	15:00 h	24 h
**HCUV + HSBS**				
Cortisol (ng/mL)	10.0 ± 1.2	4.4 ± 1.1 ^a^	4.7 ± 1.5 ^a^	1.8 ± 0.5 ^a^
α-Amylase (U/mL)	197.6 ± 37.1	283.8 ± 35.3 ^a^	302.4 ± 35.6 ^a^	239.7 ± 29.2 ^a^
DHEA (pg/mL)	301.9 ± 44.6	250.9 ± 35.3 ^a^	235.7 ± 33.6 ^a^	221.4 ± 30.3 ^a^
DHEA/Cortisol Ratio	41.2 ± 9.1	89.7 ± 17.4	107.8 ± 18.5	224.3 ± 67.2
**HCUV**				
Cortisol (ng/mL)	10.6 ± 2.3	4.9 ± 2.1	5.6 ± 2.3	2.23 ± 1.04
α-Amylase (U/mL)	231.8 ± 68.3 ^c^	317.9 ± 59.9 ^a,c^	326.2 ± 58.3 ^a^	241.0 ± 4.4 ^a^
DHEA (pg/mL)	325.1 ± 65.8	298.5 ± 50.1	264.7 ± 51.6	258.0 ± 45.5
DHEA/Cortisol Ratio	45.3 ± 16.5	102.4 ± 29.6	106.9 ± 20.4	192.7 ± 40.4
**HSBS**				
Cortisol (ng/mL)	10.4 ± 1.7	1.4 ± 0.3	4.5 ± 2.0	1.4 ± 0.3
α-Amylase (U/mL)	151.4 ± 50.4	233.7 ± 51.1 ^a^	260.9 ± 51.0 ^a^	220.2 ± 47.3 ^a^
DHEA (pg/mL)	311.2 ± 76.4	229.2 ± 61.3	233.0 ± 54.5	204.4 ± 43.4
DHEA/Cortisol Ratio	40.2 ± 13.1	85.3 ± 5.2	119.6 ± 36.1	273.8 ± 138.6
**Medical doctors**				
Cortisol (ng/mL)	10.0 ± 1.9	4.0 ± 1.9	3.0 ± 1.3	2.2 ± 1.1
α-Amylase (U/mL)	243.2 ± 68.9	318.8 ± 53.1 ^a^	367.7 ± 49.5 ^a,b^	283.2 ± 53.0 ^a^
DHEA (pg/mL)	299.4 ± 64.4	262.3 ± 65.4	245.4 ± 55.9	205.3 ± 38.3
DHEA/Cortisol Ratio	42.2 ± 13.7	102.2 ± 35.0	116.3 ± 32.2	242.5 ± 152.8
**Nurses**				
Cortisol (ng/mL)	10.1 ± 1.7	4.7 ± 1.3	5.8 ± 2.2	1.5 ± 0.3
α-Amylase (U/mL)	168.3 ± 38.1	259.4 ± 46.0 ^a^	210.2 ± 34.5 ^a^	211.9 ± 34.6 ^a^
DHEA (pg/mL)	301.6 ± 60.6	244.6 ± 40.8	233.6 ± 41.0	235.0 ± 40.0
DHEA/Cortisol Ratio	40.3 ± 11.3	80.5 ± 16.9	102.3 ± 24.1	208.1 ± 41.8
**Day shift**				
Cortisol (ng/mL)	10.5 ± 1.8	4.4 ± 1.4	5.72 ± 2.2	1.5 ± 0.3
α-Amylase (U/mL)	185.0 ± 41.8	275.7 ± 50.2	278.2 ± 52.2 ^d^	215.1 ± 36.3 ^d^
DHEA (pg/mL)	333.6 ± 67.1	273.9 ± 53.5	258.9 ± 45.9 ^d^	248.8 ± 44.2 ^d^
DHEA/Cortisol Ratio	46.4 ± 13.7	101.8 ± 27.2	122.7 ± 29.7	275.4 ± 109.1
**Afternoon shift**				
Cortisol (ng/mL)	9.6 ± 1.9	4.4 ± 1.83	3.1 ± 1.3	2.1 ± 1.0
α-Amylase (U/mL)	220.0 ± 68.1	296.9 ± 49.3	342.2 ± 44.9	280.0 ± 51.1
DHEA (pg/mL)	259.0 ± 52.8	219.6 ± 40.2	208.5 ± 50.0	185.4 ± 35.9
DHEA/Cortisol Ratio	34.1 ± 7.3	73.9 ± 14.7	88.5 ± 16.5	150.7 ± 36.0

^a^ Significant difference (*p* < 0.05) with respect to first sample obtained at the start of the working day (8:00 h). ^b^ Significant difference (*p* < 0.05) with respect to the sample obtained in nurses at 15:00 h. ^c^ Significant difference (*p* < 0.05) with respect to the sample obtained in HSBS at 8:00 h and 12:00 h. ^d^ Significant difference (*p* < 0.05) with respect to the sample obtained during afternoon shit at 15:00 h and 24 h. Abbreviations used: HCUV, Hospital Clínico Universitario de Valladolid; HSBS, Hospital Santa Bárbara de Soria.

**Table 3 ijerph-18-03937-t003:** 95% confidence intervals for the mean (using bootstrap for normalization), obtained from the MPSS-R (mean + SD) answers by the professional healthcare staff from both hospitals (HCUV, Hospital Clínico Universitario de Valladolid; HSBS, Hospital Santa Bárbara de Soria).

SCALE MPSS-R	HCUV	HSBS
Organizational stress	21.2 ± 3.9	16.3 ± 3.4
Negative patient attitudes	14.2 ± 3.0	11.6 ± 3.3
Job dissatisfaction	15.9 ± 2.9	15.0 ± 2.6
Somatic distress	20.4 ± 3.2	16.6 ± 3.1
Total stress	71.7 ± 5.6	59.5 ± 5.8 ^a^

^a^ Significant difference (*p* < 0.05) with respect to HCUV.

## Data Availability

The data that support the findings of this study are available from the corresponding author, upon reasonable request.

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
