# Peer review of "Stress Salivary Biomarkers Variation during the Work Day in Emergencies in Healthcare Professionals"

_ijerph, 2021, doi:10.3390/ijerph18083937_

Round 1
Reviewer 1 Report
This resubmission addresses some of the issues raised by reviewers but some still remain regarding the following author responses.
- Re comment 3. The title has been changed but the issue in the Abstract is that baseline values are problematic regardless of whether or not this study is about acute or chronic stress. Not having those values doesn't simply shift the focus to acute stress.
- Re: comment 8. A list of abbreviation has been provided but it would help the reader of the table heading also identifies HCUV and HSBS as two hospitals. Also, in the narrative please provide a reference that the association of DHEA/cortisol ratio is associated with cardiovascular disease.
- Re: comment 10. No mention is made that table 3 identifies a significantly lower total stress score in HSBS. Does this affect the biomarker data?
- Comment 12. Also regarding table 3 please comment further by noting the time-related correlations within each biomarker dataset and make clear that the cited correlation between cortisol and amylase is the only correlation between markers, and then only at one time point. Also the format is unconventional. Why are the data presented in reverse time ie 24h, 15h, 12h and 8h, especially as the tables are more conventionally presented?
- Re: comment 14. Page 9 line 4. Why do you suggest that there is 'ANS dysregulation'? It is interesting that amylase increased when cortisol did not in the later part of the day but could that not be adaptation rather than a failure of ANS regulation? It seems to me that the lack of cortisol when epinephrine is increased is the more interesting finding. There is published evidence that acute stress reactivity (cortisol) is decreased when someone is chronically stressed and this could support your findings of reliance on epinephrine.
- Re: comment 16. Page 9 line 13-on. Why the interest in emergency operators when the cortisol response identified when stressful call are received looks part of a normal acute stress response? Presumably you are trying to differentiate the amylase and cortisol data in this study. If so the point that chronic stress can lead to hyporeactivity of cortisol secretion would suffice.
- Re: comment 17. Returning to table 3 again, in the absence of baseline values then it would help to have an indication of published reference scores - preferably quartiles - for tool since then it might be possible to justify that staff had high stress according to the MPSS. How would we know if Table 3 reflects high or normal stress?
Author Response
Dear reviewer
I am attaching a document with the corrections to the questions you have raised.
Hope it can answer your questions
Best regards
Prof. Alfredo Córdova

Reviewer 2 Report
The authors have addressed my comments. Just needs a good review of sentence structure and wording - nothing major.
Author Response
Dear reviewer
Thank you very much for your contribution and suggestions.
Best regards
Prof. Alfredo Córdova
Reviewer 3 Report
When I accept and revise a manuscript, my objective is always, in a constructive perspective, to try to contribute so that, with my suggestions, it can be improved, when appropriate, and be published. In this sense, I thank the authors for having considered my comments valid and for accepting my suggestions. The effort made by the authors must be recognized and it seems to me that with the reservations that the authors themselves assume the manuscript can be published.
Author Response
Dear reviewer
Thank you very much for your contribution and suggestions.
Best regards
Prof. Alfredo Córdova
This manuscript is a resubmission of an earlier submission. The following is a list of the peer review reports and author responses from that submission.
Round 1
Reviewer 1 Report
Dear Authors,
I appreciated your article, is clearly exposed and very interesting.
I found similar articles, I suggest to compare your study to previous similar studies, such as
- Soo-Quee Koh D, Choon-Huat Koh G. The use of salivary biomarkers in occupational and environmental medicine. Occup Environ Med. 2007;64(3):202-210. doi:10.1136/oem.2006.026567
- Bedini S, Braun F, Weibel L, Aussedat M, Pereira B, Dutheil F. Stress and salivary cortisol in emergency medical dispatchers: a randomized shifts control trial. Plos one 2017. doi.org/10.1371/journal.pone.0177094
- Ali N, Nater UM. Salivary Alpha-Amylase as a Biomarker of Stress in Behavioral Medicine. Int J Behav Med. 2020 Jun;27(3):337-342. doi: 10.1007/s12529-019-09843-x. PMID: 31900867; PMCID: PMC7250801.
Moreover, I suggest to compare the results of the study with stress salivary markers of other samples: example medical students.
Reviewer 2 Report
This study reports an evaluation of stress in emergency medical personnel and nurses by measuring stress biomarkers in saliva. The use of English at times confuses (e.g. P1 line 1 I am unclear as to 'urgencies' - is this 'emergencies'?' Sanitaies' at the bottom of p3 looks to be an error. I am also unclear about participants being referred to as emergency staff in some places and sanitary staff in others). I am also unfamiliar with the format of Figure 3 - I did not find the presentation clear - only one correlation is noted in the text which could explain the lack of pattern discernible. Are these biomarkers normally correlated? Presentational issues apart, there are some content issues to reconsider. The measuring of biomarkers across 8.00am to 0.00h is interesting and it is unusual to see three biomarkers being assessed simultaneously. However, I think there are a number of issues, which I list here in no particular order of significance but rather as they occurred to me whilst reading the paper.
- Page 1 para 6 confused me. A clearer explanation for the biomarker secretion is needed to distinguish the HPA activation from the AMSS one. The paragraph seems to imply a differential secretion of cortisol and adrenaline whereas in acute stress both are secreted almost simultaneously - both involve brainstem activation and cortisol simply lags behind adrenaline(epinephrine) due to secretion dynamics. Little is noted regarding amylase.
- An assumption is made that stress is evident in the participants. Emergency staff do experience a lot of stress in the course of their work but we have no baseline measures here to identify if this might be acute or chronic stress in the participants. Cortisol behaves differently in chronic stress. Taking measures across the day suggests acute stress in this study but is it superimposed on a chronic stress background? We can't ascertain this from the data. Also, the cortisol diurnal pattern changes with night working and some participants worked in the morning while others were on call at night, yet they are not distinguished. Likewise no mention is made of the sleep logs noted in the Methods - did these evidence stress? Also how might the 8am value relate to the waking cortisol response?
- On P3 penultimate paragraph it is not stated how many were excluded for the reasons cited. Was this in generating the 97 participants or were they exclusions from them?
- Regarding the Findings. The circadian rhythm for cortisol looks evident and it is interesting that DHEA and amylase follow a different pattern. The changes in cortisol look normal. Would a similar change in DHEA, or amylase, be anticipated? Do they have a circadian rhythm to the extent that cortisol has? The later Discussion suggests a number of confounding variables for individuals and hence the possibility of different stress status across participants. How reliable therefore are the daily patterns? The narrative on DHEA/cortisol ratios requires referencing - how does an increased ratio relate to both anabolic and catabolic predominance, as stated in Para 1 of the Results? Or is this an error. Would DHEA be expected to be different? The difficulty I have stems from not knowing the stress status of the participants. The MPSS data exceed 50 in both hospitals, indicative of stress, but only that from HCUV looks significantly higher than 50 and possibly it is actually higher than the HSBS data. Is this the case? If so then perhaps comparing the hormone responses from the two might better support the claim of stress responses, modified or otherwise, in the personnel from HCUV.
- In the Discussion I am not sure of the rationale to suggest that the adrenal cortex has become adapted. How do the data support that? Are daytime values different to reference data (i.e. lower)?
Reviewer 3 Report
The manuscript “Stress Salivary Biomarkers Answer During The Work Day In Emergencies In Healthcare Professionals” addresses an interesting and important issue with practical implications and that can help to manage the situation of health professionals in the context of pressure/stress derived from the type and conditions of work.
However, I believe the authors should take another approach based on the design, as it seems to me that without control/baseline values, and without correlations being established, it seems difficult to say that the variations observed are responses to stress.
Comment 1: I suggest a list of abbreviations at the beginning of the article.
Comment 2: : The Abstract must be changed according to the suggestions I make, and if accepted by the authors.
Comment 3 (Title): I suggest “Stress Salivary Biomarkers Variation During The Work Day In Emergencies In Healthcare Professionals”
This suggestion has to do with all my comments related to the conception/design of the work. If reference values/controls are not presented, it is difficult to state that the results obtained are even associated with stressful situations. I now raise the question that could (should) have been explored, at least referred: acute stress vs chronic stress. Since these professionals are deployed in emergencies and are used to these conditions, what are their baseline levels for the studied markers (what would be a situation of chronic stress)? If the values were determined on days that were assigned to the emergency service, we would already be in the presence of an acute stress situation.
1. INTRODUCTION
Comment 4: “Sanitary professionals in urgencies is associated with chronic exposure to daily work stress circumstances that alter the stress hormones”.
If the authors make this statement, it is because there are works that prove it, therefore, this statement justifies a reference. Among others I suggest this and replaced “…that alter the stress hormones” by “…that alter stress biomarkers”.
Chojnowska, S.; Ptaszynska-Sarosiek, I.; Kepka, A.; Kna´s, M.; Waszkiewicz, N. Salivary Biomarkers of Stress, Anxiety and Depression. J. Clin. Med. 2021, 10, 517. https://doi.org/10.3390/jcm10030517
Comment 5: I suggest change the sequence: the first sentence seems to be good, to fit the study, then the definition of stress, then stress markers in general, then the focus on health professionals, in particular in an emergency hospital context.
Comment 6: The authors finish the Introduction with “In view of these data, we considered assessing in the present report the stress response of emergency health professionals throughout the working day, determining the levels of salivary cortisol and α-amylase, as well as the counteracting effects of DHEA”. This Comment has to do with Comment 3, because having no baseline values (which would serve as a control) I have some doubts whether we can consider responses or just variations in the markers.
2. MATERIAL AND METHODS
Comment 7: Authors must say, or at least refer to another work, how the saliva samples were treated after they were obtained, if maintained on ice and further maintained at −20 °C, until laboratory analysis, or not, and centrifugation conditions for salivettes (x min at xxx g, at xxx °C).
3. RESULTS
Comment 8: The authors present a series of categorizations [gender, skills (doctors vs nurses) work shift (66 worked in morning vs 31 during the night) and 2 Hospital], but then the data are presented all encompassed. I think the conclusions may be skewed and it would have been more interesting to compare, for example:
Doctors x Nurses
Day shift x Night shift
Hospitals (Working conditions?). As the results were presented (Table 3) we have difficulty in perceiving/establishing any correlation with the variation of the markers/possible levels of stress.
Comment 9: Please correct “Table 2 shows the total values obtained from the 3 stress-related hormones deter-mined in this study” for
“Table 2 shows the total values obtained for α-Amylase and the 2 stress-related hormones determined in this study”.
Or “Table 2 shows the total values obtained for the 3 stress-related biomarkers determined in this study”.
Comment 10: Please correct title of Table 3.
Comment 11: Please correct “Figure 3 shows the correlation chart between the 3 hormones” and the respective legend “Figure 3. Correlation chart of the 3 salivary hormones (cortisol, α-amylase and DHEA) at the different moments of the day (8:00, 12:00, 15:00 and 24 h)”.
Comment 12: The authors refer: “Figure 3 shows the correlation chart between the 3 hormones. A positive correlation (r= 0.6) appeared between cortisol levels at 24 h and α-amylase at 8:00 h. A likely expla-nation is that the decrease in cortisol at the end of the day, prepares for an increased re-sponse of α-amylase that reacts quicker than cortisol to maintain a state of stress”.
This explanation should not be here, because this is the results section. But other than that, I don't think it makes much sense. The release of amylase does not occur in order to maintain stress... amylase is released in response to stress and not to maintain it. It is a consequence and not a cause. Amylase has well-defined functions to protect the oral cavity and digestive tract. Quite possibly the rhythm in its secretion will have more to do with these functions than with "keeping stress". The circadian rhythm alone does not say anything about stress, again I mention that there should have been a control group (no stress).
DISCUSSION
Comment 13: A Discussion should be rearranged talking about cortisol, DHEA and alpha-amylase, in an orderly sequence.
Comment 14: The authors refer “This study demonstrated that the excretion of salivary proteins can change due to stress. The main observation of the present report indicates that the stress generated throughout the working day relies in the AMSS (adrenomedullar sympathetic system) response.”
I think the study doesn't show this, it just shows that there is a variation of some of these analytes throughout the day. As the levels of amylase were not correlated with the levels of stress obtained by the questionnaires, we cannot say this.
Comment 15: The authors refer “The simulation reported increases in α-amylase, similar to those observed in a real work scenario (present report)”.
Same as in Comment 14.
Comment 16: The authors refer “However, it has been suggested that the α-amylase is mostly a calming or relaxation marker”.
If so, a bibliographic reference must be provided.
Comment 17: The authors refer “The study showed that stress in teams of doctors and nurses is a real fact”.
I still have my doubts that with this design and without baseline values/ controls the authors can make this statement.
Reviewer 4 Report
The study provides preliminary data on levels of stress of emergency room health care providers. Yet there are some reasons for stress not accounted for in the method used.
Translation errors are noted throughout the manuscript.
Use of legends on figures encouraged rather than including information in the figure description. For example - using the colors for men and women as legend.
More information on method is needed:
Were the subjects working at the time of data collection? If working the "morning shift" then not all 4 collection times were completed while working.
What are the hours of the "morning shift" - was 8 AM 2 hours into work day? etc.
Data on sleep was collected on 5 days - no information on how this relates with the data?
What was the years of experience in this environment and in role overall for the subjects. Noted in the literature that experience can make a difference in performance.
How did the data from the perceived level of stress related to the saliva data?
Description of the process for collection of the saliva data is needed. Authors indicate the subjects were notified 30 minutes before obtaining sample – how? In the discussion it sounds like a researcher collected the data but this is not addressed in the method section.
If the data were collected during work time addressing what the perceived level of stress just prior to the saliva collection would have been meaningful. For example – having just admitted someone with a major injury could be perceived differently than treating a patient with a cough. Could change the levels.
Authors indicate that some of the subjects do call during the night shift. Was this true for any of the subjects on the data collection day – as well as prior to the day of data collection.